# The Size Effect on Flexural Fracture of Polyolefin Fibre-Reinforced Concrete

**Álvaro Picazo** [1] , **Marcos G. Alberti** [2] , **Jaime C. Gálvez** [2,*] , **Alejandro Enfedaque** [2] **and Abner C. Vega** [2]

1 Departamento de Tecnología de la Edificación, E.T.S de Edificación, Universidad Politécnica de Madrid, Avda. Juan de Herrera, 6, 28040 Madrid, Spain; a.picazo@upm.es
2 Departamento de Ingeniería Civil: Construcción, E.T.S de Ingenieros de Caminos, Canales y Puertos, Universidad Politécnica de Madrid, C/Profesor Aranguren, s/n, 28040 Madrid, Spain; marcos.garcia@upm.es (M.G.A.); alejandro.enfedaque@upm.es (A.E.); abnercivil@gmail.com (A.C.V.)
* Correspondence: jaime.galvez@upm.es; Tel.: +34-91-0674-125

**Abstract:** The reinforcement of concrete by using polyolefin fibres may be considered in structural design to meet the requirements of the applicable code rules. To achieve a reliable use of such a composite material, use of full-scale real structures is needed. The conversion of lab testing data into real practice properties is challenging and significantly influenced by various aspects, among which the size effect is a key one. Given that the available literature does not report coinciding conclusions about such an effect on quasi-brittle materials reinforced with fibres, further research is justified. Therefore, this work studies the behaviour of notched beams with three proportional sizes by using self-compacting polyolefin reinforced concrete with a fibre volume fraction of 1.1%. Flexural testing was carried out according to the standard EN-14651, with the results revealing the existence of the size effect. In addition, a reduction of the residual strength identified in the larger specimens was observed in fracture surfaces with equal fibre content.

**Keywords:** polyolefin fibre-reinforced concrete; fracture behaviour; size effect; bending tests

## 1. Introduction

Concrete is the most extensively used material in building and civil construction [1]. Although as a structural element it exhibits excellent behaviour for compressive actions, it lacks sufficient resistance for tension loadings and behaves as a quasi-fragile material does [2]. To address this weakness, steel reinforcing embedded bars or, more recently, fibres randomly distributed in the concrete mass have been used. In the steel-bar option, commonly called reinforced concrete, bars assume the tension stresses that may appear in the member sections. As it improves crack control and the effect of concrete shrinkage [3], the option of fibre reinforcement (fibre-reinforced concrete, FRC) may be a partial or total substitution of the steel bars as long as they have been considered in the design calculations [4,5], complying with the requirements of the pertinent structural design codes [6,7]. Nonetheless, both reinforcement options make the concrete behave as a ductile material does.

The most popular types of fibres used nowadays for structural concrete are made of steel or polyolefin. Given that use started in the mid-20th century [8], steel fibres have been widely studied with a significant amount of research work being reported to assess the relevant properties [9]. However, use of polyolefin fibres started only in the late 20th century as a result of the development of the chemical industry. Given that the lack of experience in use of this fibre type has led to a relatively low awareness of the fibre mechanical properties, deeper understanding of use of such a fibre type when it is combined with concrete is required.

A short comparison of steel and polyolefin fibres shows that the latter improve certain aspects that should be considered when the choice of one or the other is under discussion: the absence of oxide deterioration [10], lack of sensitivity of magnetic fields, absence of safety problems due to protruding fibres at the element surface [11] and lower wear of machinery for the fabrication, transport and placing of concrete when compared with use of steel fibres [12,13].

One of the problems for transferring the research findings to the industrial applications of new techniques or materials is that the direct use of the laboratory conclusions and the required formulation parameters for member design is not always possible. In a quasi-brittle material such as concrete, it was observed that when the size of the specimen increased and the shape proportions maintained, the level of collapse stresses (that is to say, the specimen strength) decreased. This is known as the size effect [14].

Regarding such an effect, it is important to note that there are three theories involved in its study: statistical, deterministic and that based on fractals [15], with the deterministic one being the most used [16]. The size effect is defined as the deviation of the real strength capacity compared with the load predicted by plastic analysis or another classical structural strength theory based on critical stress states [17]. This effect can be graphically represented, as in Figure 1, being at least three sizes necessary in order to fit the size effect law of the material.

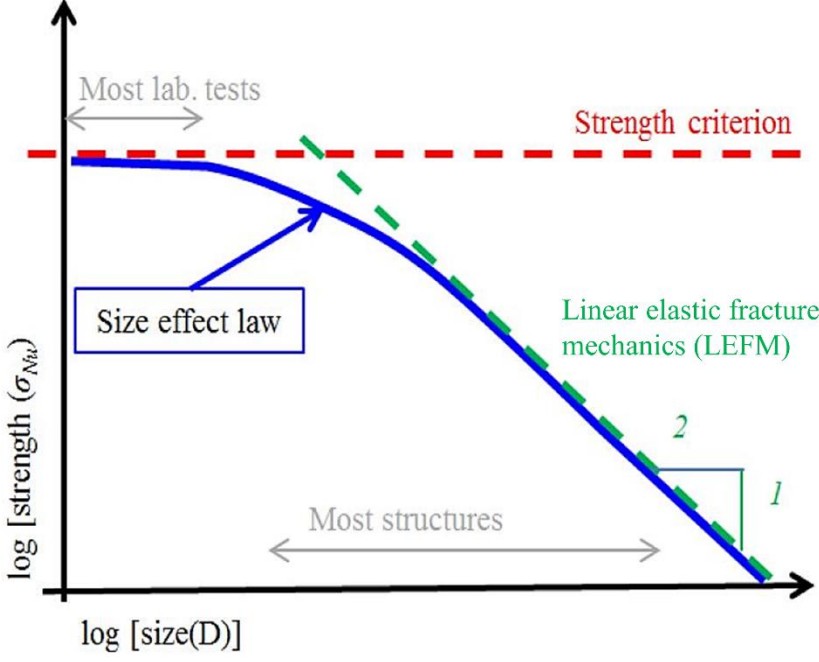

**Figure 1.** Size effect on member strength (adapted from Reference [18]).

Although the existence of the size effect is widely accepted by the scientific community, there remain discrepancies regarding assessment in FRC. Some research campaigns have been carried out with regard to the influence of the maximum aggregate size and the added fibre volume fraction, with studies revealing the existence of a size effect in such a concrete [19]. Fracture and brittleness data was obtained in the research performed by Bažant [14]. In addition, there is also published research which concludes that fibre-reinforced concrete with a high content of ductile steel fibres shows an almost negligible size effect [20]. This conclusion is consistent with other papers which argue that with high-volume fibre content the size effect is small due to the increment of ductility provided by the fibres [21]. Lastly, other reports that conclude that the size effect is more important in high-strength concrete should be mentioned [22].

Therefore, this lack of consensus on the influence of the size effect in fibre-reinforced concrete justifies the need of the present research which analyses the residual strength provided by the reinforcing

fibres. The residual strength also depends on certain other factors considered: the fibre–concrete interaction, fibre orientation and distribution (the orientation factor [23]), and the wall effect [24].

A previously concrete component formulation was used [25] to produce self-compacting concrete with a fibre volume fraction of 1.1%. Polyolefin fibres 48 mm long were used. With this material, beams were cast in distinct sizes with the same dimension proportions and, after being notched, subjected to a three-point bending test. With this testing campaign, the existence of the size effect in polyolefin fibre-reinforced concrete was confirmed, with a decrease in the residual strength being detected in the larger specimens with the same amount of fibres in the fracture surface.

## 2. Experimental Programme

The experimental campaign entailed the casting of beams in three sizes made of self-compacting polyolefin fibre-reinforced concrete containing 10 kg/m$^3$ of 48 mm long polyolefin fibres. All beam dimensions were defined as proportional to the beam depth and the proportions were equal in all specimens. The diversity of sizes and amount of specimens allowed enough data to be gathered to analyse the results. The materials used and information about how the beams were tested are described below.

### 2.1. Materials and Specimen Fabrication

Portland cement type EN 197-1 CEM I 52.5 R-SR5 (ManuCEMEX, Castillejo, Madrid, Spain) was used in the concrete mix called SCC10. The cement was mixed with limestone filler with a calcium carbonate content higher than 98%, a 45 μm sieve retention of less than 0.05%, a density of 2700 kg/m$^3$ and a Blaine surface of 400–450 m$^2$/kg. The silica aggregate contained two coarse size ranges: 4–8 mm grit and 8–12 mm gravel with a maximum size of 12.7 mm. The sand size range was 0–2 mm. Polycarboxilate-based superplasticizer, named Sika-Viscocrete 5720 (ManuSIKA, Madrid, Spain), with a density of 1090 kg/m$^3$ and a solid content of 36% was also included in the mix. The mix formulation is shown in Table 1 and was that used in previous experimental campaigns [25]. The 48 mm long macro fibres type SikaFiber, with an embossed surface, were added to the mix in a volume fraction of 1.1%, equivalent to 10 kg/m$^3$. Table 1 also shows the main fibre properties. The visual aspect of the fibres can be seen in Figure 2.

**Table 1.** Concrete composition and fibre properties.

| Material | SCC10 | PF48 Fibre Properties | |
|---|---|---|---|
| Cement (kg/m$^3$) | 375 | Length (mm) | 48 |
| Limestone (kg/m$^3$) | 200 | Equivalent diameter (mm) | 0.903 |
| Water (kg/m$^3$) | 188 | Aspect ratio | 53 |
| Water/cement | 0.5 | Tensile strength (MPa) | >400 |
| Gravel (kg/m$^3$) | 245 | Density (g/cm$^3$) | 0.91 |
| Grit (kg/m$^3$) | 367 | Modulus of elasticity (GPa) | >6 |
| Sand (kg/m$^3$) | 918 | Fibre shape | Straight |
| Superplasticizer (% cement) | 1.25 | Surface structure | Rough |
| PF48 (kg/m$^3$) | 10 | Fibres per kg | 32,895 |
| Fibre volume fraction (%) | 1.10 | | |

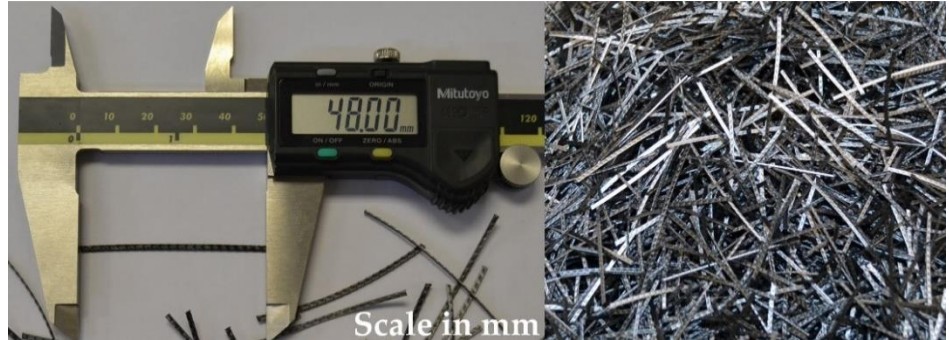

**Figure 2.** Visual aspect of the polyolefin fibres.

A 100-litre capacity vertical axis mixer was used in the concrete fabrication. The mixing sequence was as follows. First, the three types of aggregate were mixed for one minute, followed by the addition of 1/3 of the fibre content which was mixed for 30 more seconds. Then, cement and filler were added and mixed for another 30 s. After that, 30 more mixing seconds were needed before the incorporation of the other 1/3 of fibre content. Next, after 75% of water had been poured and mixed for another minute the rest of the water, the fibres and the superplasticizer were added, continuing the mixing action for 150 s. Then the mixer was stopped for 150 s. The mixing was then resumed for two minutes.

Fresh concrete was tested in a slump-flow test according to the standard procedures [26]. The double-test average results were $t_{500}$ = 6 s and the diameter of the patty $d_m$ = 570 mm. Concrete was poured into the moulds from one end, leaving the concrete flows towards the opposite end with the only compaction action being its own weight as it is recommended by the standards and bibliography [24,27]. The fresh specimens were covered with plastic film to keep the upper surface from drying. The beams were unmoulded 24 h later and placed in a humid chamber at 20 °C and 90% of relative humidity for a minimum of 28 days (the point when they were ready for testing).

Before the bending tests, a half-depth cut was machined in the lower part of the beam central sections with a low-speed diamond disc, carefully maintaining the integrity and dimension of the upper half section of the specimen. The proportional dimensions based on the specimen depth $D$ are shown in Figure 3. Under this procedure, three specimen size categories were obtained: large, medium and small, as defined in Table 2. Figure 4 shows a photo with the three sizes of the tested specimens.

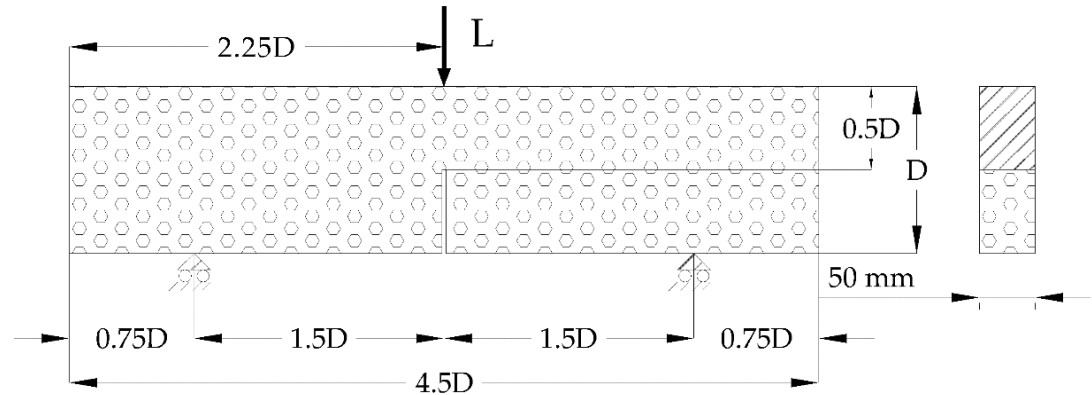

**Figure 3.** Specimen dimensional proportions and testing support and loading setup.

**Table 2.** Size and number of valid testing specimens.

| Specimen | Units | Length (mm) | Width (mm) | High (mm) |
|----------|-------|-------------|------------|-----------|
| Large    | 3     | 1350        | 50         | 300       |
| Medium   | 2     | 675         | 50         | 150       |
| Small    | 3     | 340         | 50         | 75        |

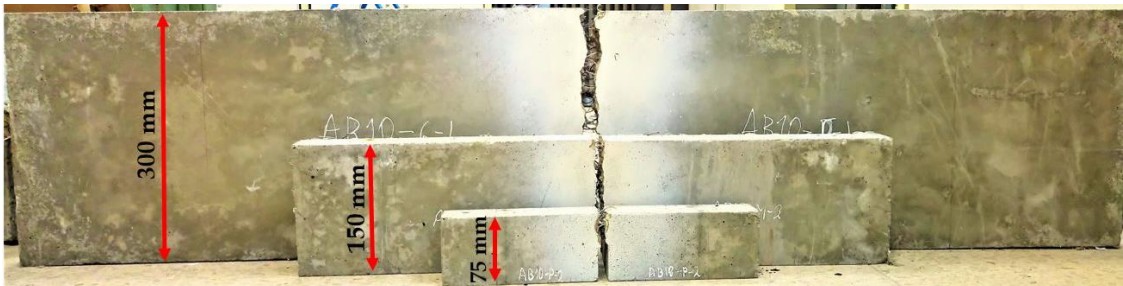

**Figure 4.** Photo of the three sizes of the tested specimens.

*2.2. Testing Development*

Once the specimens were ready for testing, the dimensions were checked in order to locate the positions of supports, loading cylinder and notching cuts. The surrounding areas of the cuts on both sides of the specimens were painted with a random point monochrome [28] template in order to apply digital image correlation (DIC) techniques. Crack generation and further development were detected by the position data of the template points during the test time. The specimens were then ready for bending testing according to standard EN-14651 (apart from the specimen sizes) [27]. That is to say, the test method in the standard was used to assess the fracture behaviour of different specimen sizes than that established by the standard ($600 \times 150 \times 150$ mm$^3$). The main testing equipment was one servo-control Instron 8803 press machine (Instron, Norwood, MA, USA) with a maximum loading capacity of 500 kN and a 25 kN Dynacell TM loading cell. A crack mouth opening displacement (CMOD) device (Instron 2620-602, Norwood, MA, USA), with a range of up to 8 mm, was used to measure the relative opening of the notching cut lower tips. The beam deflections were controlled by two linear variable differential transformer (LVDT) devices (Instron 2601-044, Norwood, MA, USA) with 30 mm range, placed on either side of the beam. The final deflection value was the average of those of both devices. Furthermore, to verify the time development of the cracks two five-megapixel high-definition cameras were installed on either side of the testing arrangement for detecting and monitoring the displacements of points in the area where the cracks would presumably start. Recording frequency was set to one picture per second so that the pictures were synchronised with the testing machine.

The specimens were placed on the supports (as shown in Figure 3) and then the CMOD and LVDT devices were positioned by using a laser levelling instrument to ensure beam horizontality and that the loading cylinder was acting in the middle of the span (as shown in Figure 5). Lastly, the cameras were placed in position and activated on each side of the test configuration.

The actuator displacement was used for loading control at an initial rate of 0.6 mm/min and in a second phase at 0.17 mm/min. In each test the first crack appeared in the first phase and the end of testing occurred in the second phase.

The positioning of fibres within the concrete matrix plays a significant role in the strength capacity of the lab specimen and the real structural element and its reliability [29]. A counting procedure of the existing fibres in the fracture surface was carried out based on dividing the section area in eight zones, as shown in Figure 6 in order to consider the main anisotropy effects such as the wall effect due to the formwork sizes. These zones are generated by using a band width of 24 mm (half of the fibre length) around the mould perimeter. The wall effect disturbs the fibre positioning and affects the fibre distribution and orientation. The wall effect tends to orientate the fibres along the surface of the wall and it is accepted in the literature that it affects the surfaces closer than half the fibre length to the mould walls, as shown in Figure 6.

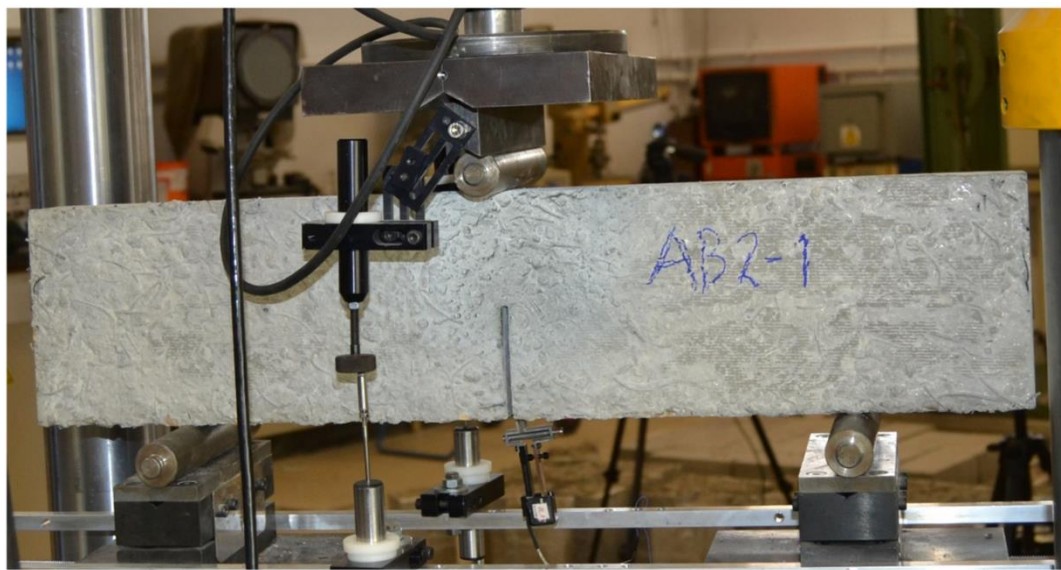

**Figure 5.** Beam specimen ready for testing.

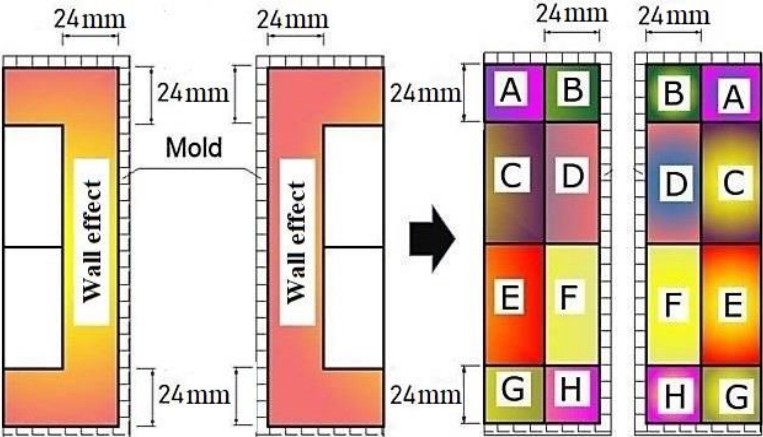

**Figure 6.** Mould wall effect and zones for fibre counting in the fracture surface.

## 3. Results

The following testing data were: elapsed time, load, actuator displacement, crack mouth opening displacement, beam deflection and video images of the crack generation and development. Each, in combination with the fracture surface analysis, was used to prepare the result report. The load and stresses occurred in the test time along with the beam deflections and the notch opening values were shown. In addition, the relationship between the specimen size, the number and orientation of fibres in the fracture surface was also analysed.

The residual strength of the specimen depends on the applied load, the beam span and the resisting section properties, and is assessed by means of Equation (1), proposed by standard EN-14651 [27]. In Equation (1) $F_j$ is the applied load, $L$ is the span between supports, $b$ the specimen width and $h_{sp}$ the length of the ligament resisting section.

$$f_{R,j} = 3F_jL/2bh_{sp}^2 \tag{1}$$

### 3.1. Fracture Behaviour Results

The first point observed in the tests was the evolution of the CMOD versus the applied load. Figure 7a shows the average results for each of the three specimen sizes. Two main turning points

may be observed in the plots defined by the behaviour changes occurred in the test time. Initially, a steep line appears up to the maximum load. In this test, part the connecting section in the beam centre remained unaltered. The final point coincided with the appearance of cracks in the resisting section followed by a sharp drop of the load down to a minimum value, the second critical turning where the fibres started reloading. The energy absorbed by fibres generates the post-cracking curve branch shown in the plots. In the medium-size specimens a third spot or turning point can be observed for a CMOD value of approximately 4.5 mm, which gives rise to a slightly decreasing curve of the residual load down to the end of the test. This is the expected behaviour of polyolefin fibre-reinforced concrete (PFRC) as shown in references [13,24,25,29].

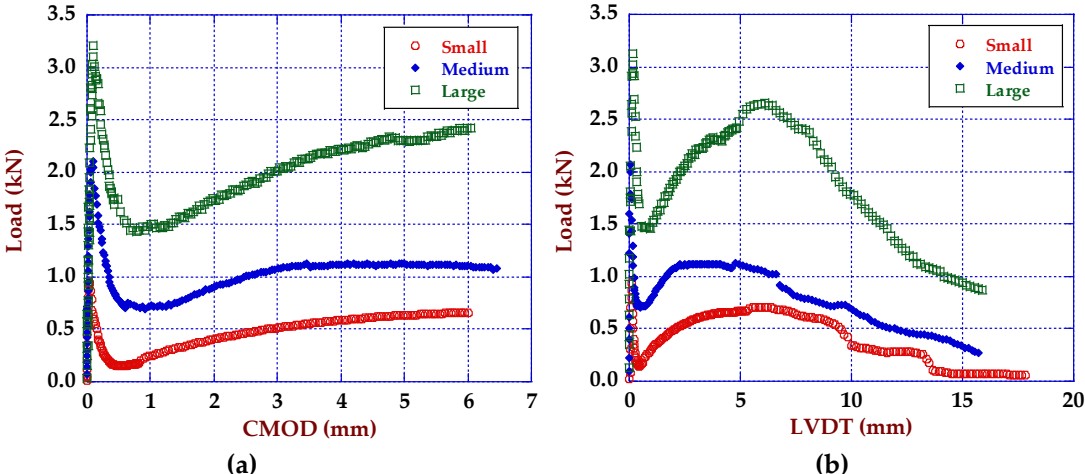

**Figure 7.** Average fracture curves for the three specimen sizes: (**a**) load-crack mouth opening displacement (CMOD) plots; (**b**) load-mid-span deflection (LVDT).

Similarly, Figure 7b shows the average load versus mid-span deflection (LVDT) curves for the three specimen sizes. Again, curves represent the typical behaviour of PFRC: first, a branch up to a maximum load value at the proportionality limit, $L_{LOP}$, then after the first crack is generated there is a sharply descendent branch down to a minimum post-cracking load value, $L_{MIN,}$ followed by a reloading branch when the fibres start assuming the internal tension stresses up to a relative maximum post-cracking load ($L_{REM}$) and lastly descending branch where the fibres are either breaking or sliding inside the concrete matrix. This reloading branch remained active up to a deflection value of about 5 mm. It is worth noting that all specimens maintained some residual load for deflections as high as 15 mm.

*3.2. Fracture Surface Analysis*

To assess the number of fibres that cross the fracture surface, the specimens tested, shown in Table 2, were split in two half parts through the notch cut and the fibres counted in both parts. Figure 8 shows the two halves of one specimen prepared for the fibre counting.

The orientation factor can be obtained by counting the number of fibres in the fracture surface and computing the theoretical (*th*) number of fibres that cross the section by means of Equation (2). Such value is the number of fibres that would have been counted in an idealistic positioning.

The orientation factor ($\theta$) can be defined as the relation of the number of fibres counted in a certain surface over its theoretical number. That is to say, it is possible to obtain the orientation factor once it has been counted by using Equation (3).

$$th = \frac{A \cdot V_f}{A_f} \qquad (2)$$

$$\theta = \frac{n}{th} = n\frac{A_f}{V_f A} \tag{3}$$

with $A$ being the cross section of the sample, $A_f$ the section of one fibre, $V_f$ the fibre volume fraction and $n$ the number of fibres actually counted in situ.

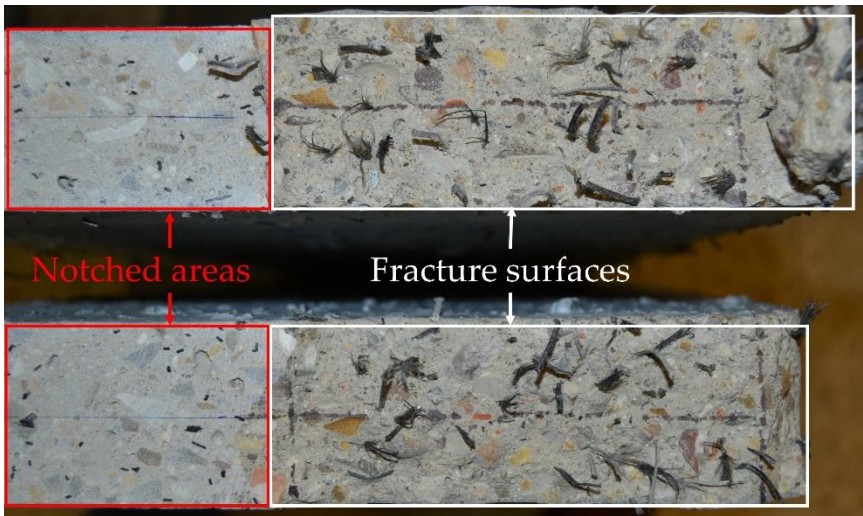

**Figure 8.** Fracture surfaces of one specimen prepared for the fibre counting.

In Table 3 the average numbers of fibres is shown, along with the variation coefficients and the global orientation factor, for each of the three specimen sizes.

**Table 3.** Average number of fibres in the fracture surface with their coefficient of variation (c.v.) and the overall orientation factor ($\theta$) for each specimen size.

| Specimen | Fibres (c.v.) | Orientation Factor ($\theta$) |
|---|---|---|
| Large | 178 (0.10) | 0.63 |
| Medium | 106 (0.03) | 0.62 |
| Small | 57 (0.08) | 0.72 |

The orientation factor is a powerful tool for analysing and predicting reliability of structural use of FRC. In this case, as can be seen in Table 3, the orientation factor reached values that exceeded 0.60 (this was expected, given the results offered in the published literature [23,29,30]). The wall effect in the case of the small specimen affected more than half of the total fracture surface. Therefore, this increased the value of $\theta$ of the small specimens if compared with large and medium ones. In addition, standard specimens prescribed by EN-14651 [27], also show improvements in the isotropic areas given that the flux of self-compacting concrete tends to align the fibres perpendicular to the fracture surface. This helps understanding of why $\theta$ is greater than 0.60. Moreover, previous research [13,31] showed that the longer the pouring distance of self-compacting concrete, the higher is the influence of the flux. That is to say, the values of the orientation factor of the large specimens were expected to be higher than the medium ones. In addition, the wall effect was evident on the orientation factor of the small specimens, with all of them being higher than a regular specimen size with a regular vibrated concrete.

## 4. Discussion

Once the data had been collected and processed, a deeper discussion and analysis of the findings was needed in order to assess the size effect in the fracture mode I of PFRC.

### 4.1. Size Effect and Strength-CMOD Curves

The curves of the average strength-CMOD curves for the three tested sizes, as shown in Figure 7, reveal clear behaviour differences. The residual strength obtained by using Equation (1), which is directly derived from the classical theory of strength of materials, already considers the size of the specimen for the calculation of the stress in the lower fibre of the ligament section. Therefore, the curves shown in Figure 9 permit a direct comparison that was not possible by only observing Figure 7. This is of high relevance because it enables a comparison of residual strengths such as $f_{R1}$ and $f_{R3}$ which are the main values of strength used in structural design.

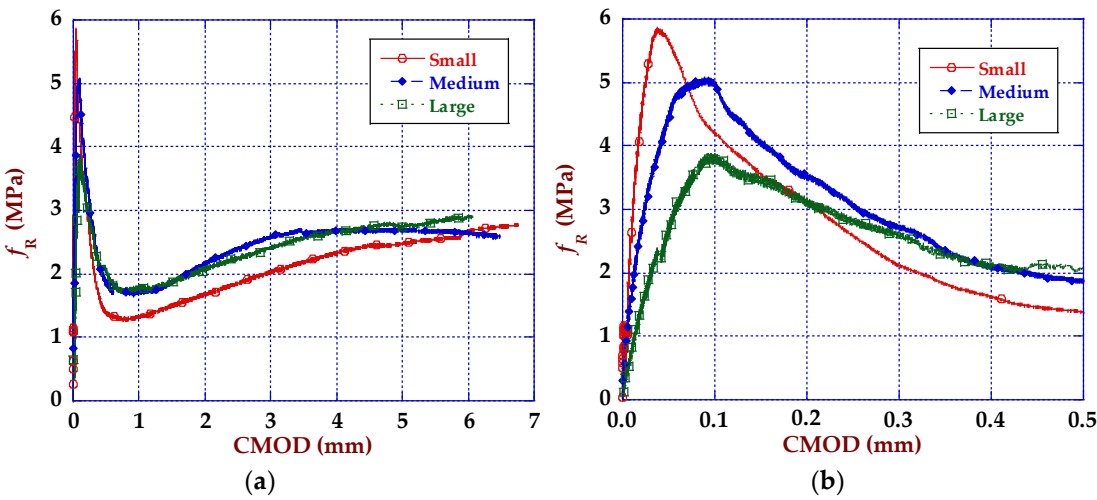

**Figure 9.** Residual strength-CMOD curves: (**a**) complete test; (**b**) for a notch tip opening up to 0.5 mm.

Starting with the strength at the limit of proportionality ($f_{LOP}$), the curve of the small specimens presents a higher peak-stress level compared with the other sizes, as can be understood by observing Figure 9. The strength values at the limit of proportionality were 5.86 MPa, 5.07 MPa and 3.85 MPa for, respectively, the small, medium and large specimens. Therefore, the small specimens experienced stresses 52.2% higher than the large size specimens (the scale effect). In such a sense, the expected following size effect classical law for concrete occurs: the larger the size, the lower are the strengths obtained. Since the limit of proportionality ($f_{LOP}$) is controlled by the concrete matrix, not by the fibres, the size effect observed is in accordance with the results of previous published works for plain concrete, such as [32].

The slopes of the first loading branch for each of the sizes were also evident, as shown in Figure 9b. The highest slope obtained in the tests of the small specimens showed a stiffer behaviour than those obtained with the medium and larger specimens. By the end of the discharge branch, for a CMOD of 0.42 mm, the strengths show similar values and are in the same order as the sizes because the strength is assumed by the fibres. This behaviour reveals the ductile behaviour of a fibre-reinforced concrete structural member.

As suggested before, the strength values at the proportionality limit ($f_{LOP}$) trend to decrease as a consequence of the specimen size increase and behave inversely to size. Contrarily, the rest of residual strengths behaved in the same direct sense as the fracture surface size, the number of fibres and the specimen size. In Figure 10 these tendencies can be observed for the three sizes with respect to the proportionality limit ($f_{LOP}$), minimum strength value at the end of the discharge branch ($f_{MIN}$) and the strengths for the CMOD values of 0.5 mm and 2.5 mm ($f_{R1}$ and $f_{R3}$) in relation to specimen depth $D$. It was then necessary to analyse the effect of the fibre counting and orientation on the residual strength in order to decouple these effects from the size effect.

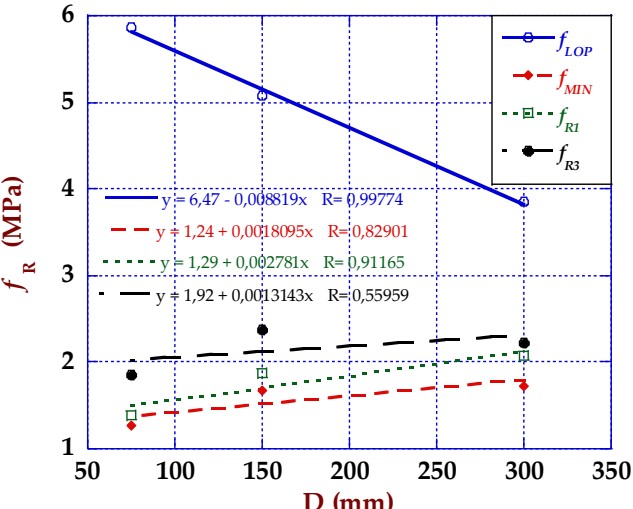

**Figure 10.** Plots of average values of proportionality limit strength ($f_{LOP}$), minimum strength at the end of discharging branch ($f_{MIN}$), CMOD opening of 0.5 mm ($f_{R1}$) and CMOD of 2.5 mm ($f_{R3}$) versus the total specimen depth *D*.

For the evaluation of the size effect on PFRC, the analysis of the residual strength is a key point. In Figure 11 the tendency lines of the residual strength values in relation to the number of fibres encountered in the fracture surface are shown. By observing Figure 11, a clear tendency of steep slope in the small specimens and a smoother slope in the medium and large specimens can be stated. It is worth mentioning that as only one fibre dosage was used, the number of fibres remained for each size in close values.

In such a way, starting at the point of minimum residual strength beyond the first discharge branch ($f_{MIN}$) the curve shows increasing strength values which mean an increase in the fracture energy up to a second relative maximum ($f_{REM}$) and a smoothly decreasing residual strength. This behaviour shows the superior ductility and toughness of PFRC compared with unreinforced concrete.

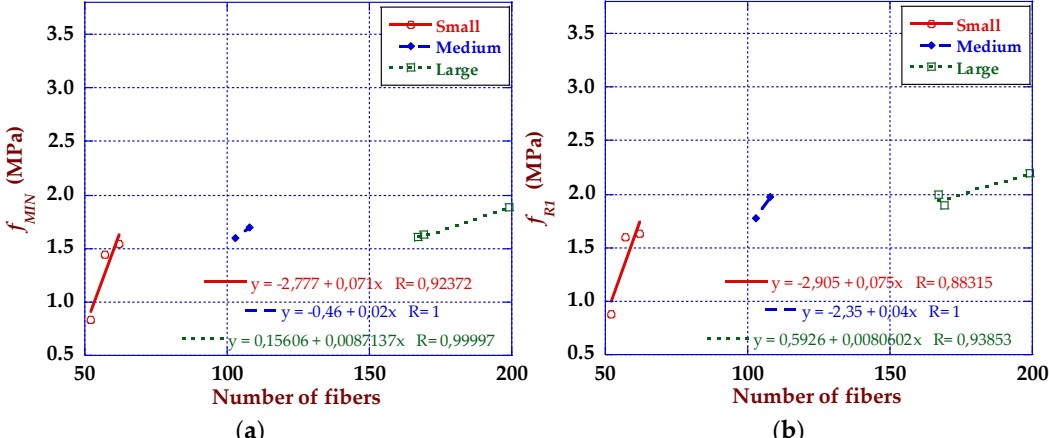

**Figure 11.** *Cont.*

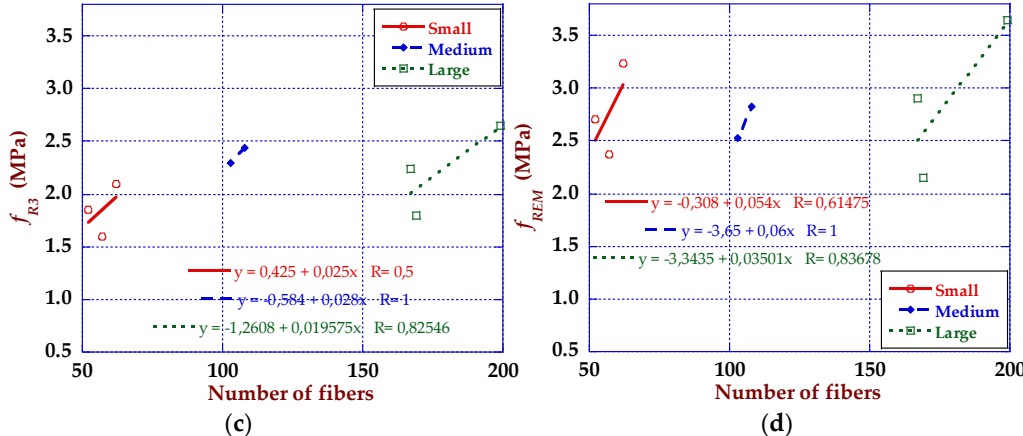

**Figure 11.** Tendency lines of the distinct residual strength values in relation to the number of fibres for the three specimen sizes: (**a**) for the minimum strength ($f_{MIN}$), (**b**) for CMOD of 0.5 mm ($f_{R1}$), (**c**) for CMOD of 2.5 mm ($f_{R3}$) and (**d**) for the maximum residual post-cracking strength ($f_{REM}$).

The analysis of the relationship of residual strengths versus number of fibres also confirms the existence of the size effect.

Figure 12 was prepared to analyse the size effect by using the plot residual strength versus number of fibres which shows the superior behaviour in terms of strength of the small specimens compared with the larger sizes. Considering all the sizes and fitting the lines passing through the origin, the tendency of each size could be found. If the figure is observed in detail, for a specific number of fibres in the fracture surface the strength obtained follows the size effect expected tendency. That is to say, the larger the size the lower is the strength for the same number of fibres. In such a way, the size effect existence was observed by decoupling the influence of the fibre distribution on the fracture surface.

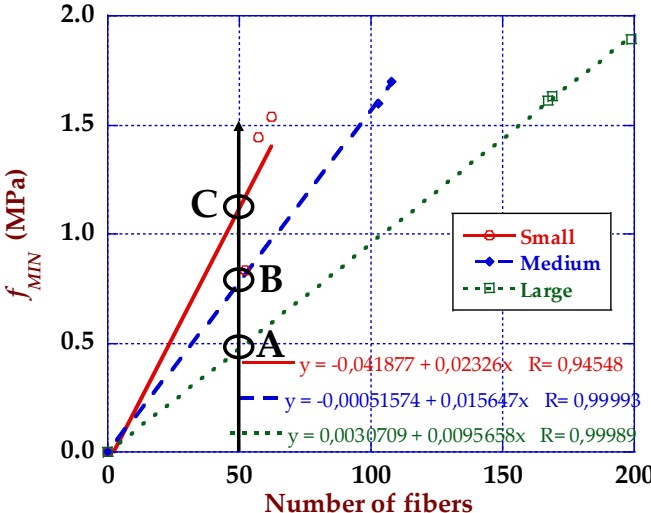

**Figure 12.** Fitting lines of the relationship average minimum residual strength versus number of fibres.

Figure 12 shows the values of $f_{R1}$, $f_{R3}$, $f_{min}$ and $f_{REM}$ as a function of the number of fibres in the fracture surface. As can be seen, keeping constant the volume fraction, the number of fibres in the fracture surface in the smallest specimens is about 50, in the medium-sized specimens around 100 and close to 200 for the largest specimens. To assess the size effect in $f_{min}$, Figure 12 was performed. In this figure, the results of Figure 11a were extended by a linear fitting for the comparison with the same number of fibres. It has been shown [33] that residual strength in fibre-reinforced concrete is directly related with the number of fibres in the fracture surface. Figure 12 shows that with the same number of fibres there is a clear size effect. Considering, for example, 50 fibres the strength of the large specimens

(point A) corresponds to a minimum value of 0.48 MPa, whereas this value increases to 0.78 MPa (point B) for medium-size specimens and reaches 1.11 MPa (point C) in case of small specimens. This data means a 231% strength increment in the small specimens in relation to the big specimens. Again, this behaviour confirmed that the size effect occurred for PFRC elements, according to the classical theory of size effect for quasi-brittle materials [32].

With this procedure it was possible to conclude that the size effect took place not only in the strength values depending on the concrete matrix ($f_{LOP}$) but also in the post-cracking strength values.

## 4.2. Fracture Energy

To quantify and compare the work of fracture for each specimen size, the specific fracture energy ($G_F$) of all the mixes was analysed by means of the load-deflection curves and processed by using Equation (4).

$$G_F = \frac{W_f}{b \cdot h_{sp}} \tag{4}$$

Where $W_f$ was the fracture work borne by the sample, $b$ the width of the sample (50 mm) and $h_{sp}$ the length of the ligament (0.5$D$ for each size).

Fracture energy was assessed for several deflections, as is shown in Table 4. For deflections up to 5 mm, such fracture energy and the fracture surface depth are related. For higher deflections, the medium-size specimens absorb less energy in a process that depends on the fibre deformation and the small and large specimens maintain the tendency of having higher fracture energy with a higher fracture surface.

**Table 4.** Fracture energy values (N/m) for different deflection values and the coefficient of variation.

| Specimen | $G_F$ (1 mm) | $G_F$ (5 mm) | $G_F$ (10 mm) | $G_F$ (15 mm) |
|----------|--------------|--------------|---------------|---------------|
| Large    | 249 (0.06)   | 1436 (0.21)  | 2979 (0.21)   | 3939 (0.15)   |
| Medium   | 242 (0.01)   | 1374 (0.01)  | 2495 (0.01)   | 2722 (0.05)   |
| Small    | 163 (0.25)   | 1296 (0.18)  | 2947 (0.18)   | 3330 (0.17)   |

## 4.3. Video-Extensometry

The generation and development of cracks were studied by means of digital image correlation (DIC). That is to say, a correlation of images synchronised with the testing machine data record allows the use of the images as a video-extensometer. To perform the DIC analyses, the images were recorded at one frame per second. Such a frame rate allowed the synchronisation of the videos and the data acquired by testing machine gauges. As previously mentioned, the testing machine recorded load, displacement of the actuator, CMOD in both sides and test elapsed time. Therefore, time was common for both systems and the fixed frame rate allowed the synchronisation of the videos with the testing machine and made it possible to correlate by using the software Vic-2D. The scheme of the test and disposition of the devices can be seen in Figure 13. This study allowed the relationship of the load values produced by the actuator displacement with the crack formation images at a given test elapsed time to be established. In each case, the crack generation started in the upper point of the notch cut. The cracks progressed with a vertical trajectory clearly influenced by the fibre bridging effect. Figure 14 is the image of one of the cracks generated in the bending test.

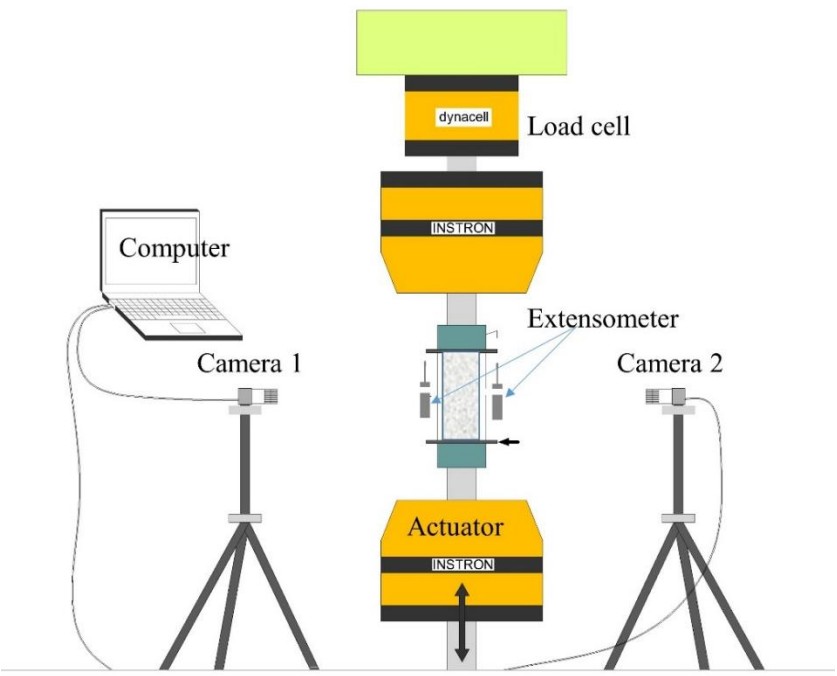

**Figure 13.** Bending test setup including digital image correlation (DIC) positioning.

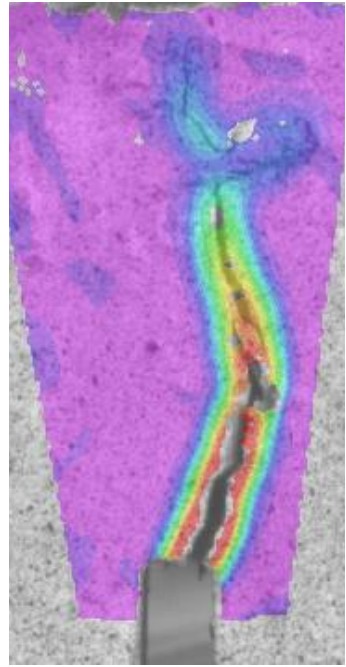

**Figure 14.** Crack generated in one of the specimens during the bending test.

## 5. Concluding Remarks

Specimens of three proportional sizes were fabricated in order to study the size effect in polyolefin fibre-reinforced concrete under fracture testing. The three-point bending tests were performed in accordance with the procedures described in standard EN-14651, adapted to the specimen real dimensions. The conventional measurement devices were improved by the installation of DIC techniques that allowed the inspection of the generation and development of cracks.

It was shown that the residual strength at the proportionality limit is governed by the composite (concrete matrix) properties and that post-cracking residual strength is governed by the fibres, meaning that the section with the highest number of fibres presented the greater strength.

A significant growth in fracture energy was observed after the initiation of cracking in each tested specimen due to the energy taken by the fibre deformation. This feature showed the ductile behaviour of the polyolefin fibre-reinforced concrete.

It was found that at the limit of proportionality, specimens with a smaller fracture surface led to higher values of strength. Since the limit of proportionality is governed by the concrete matrix, such a result agrees with classical size effect theory for plain concrete

In the case of residual strengths, the decoupling of the strengths and the number of fibres placed in the fracture surface allowed the conclusion that the size effect was taking place, detecting the reduction in residual strength in larger specimens with equal number of fibres placed in the fracture surface. This conclusion could guide the reader in examining the results and conclusions offered by previously published research that deals with the size effect and fibre-reinforced concrete.

The digital image correlation technique showed that the cracks were generated at the top of the notch cut and that they progressed upwards in a curved shape.

**Author Contributions:** Conceptualization, J.C.G.; methodology, M.G.A. and A.E.; software, Á.P. and A.C.V.; validation, Á.P. and A.C.V.; formal analysis Á.P. and A.C.V.; investigation, all authors; resources, J.C.G.; data curation, all authors; writing—original draft preparation, M.G.A. and Á.P.; writing—review and editing, J.C.G. and A.E.; visualization, all authors; supervision, J.C.G.; project administration, J.C.G.; funding acquisition, all authors under Research Fund Project BIA2016-78742-C2-2-R.

**Funding:** This research was funded by the Ministry of Economy, Industry and Competitiveness of Spain, BIA2016-78742-C2-2-R. The authors also offer their gratitude to SIKA SAU for supplying the polyolefin fibres.

**Conflicts of Interest:** The authors declare no conflicts of interest.

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
