# Peer review of "The Size Effect on Flexural Fracture of Polyolefin Fibre-Reinforced Concrete"

_applsci, doi:10.3390/app9091762_

Reviewer 1 Report

This article deals with a very interesting experimental work, study the behaviour on beams (with three proportional sizes) and what the effects of the size effect in polyolefin fibre reinforced concrete under fracture testing. The paper provides new results and contributes to improve the state of the art in this line of research. It is considered that this paper can be published.

Is well written and easy to read.

Few improvements are needed. Please see the following comments aiming at improving the paper.

Figure 1. explain the he meaning LEFM?

Figure 2. decrease the scale, for example 1/50
Line 140 - what was the standard used for the manufacture of the specimens. In the test is referred that “The specimens were then ready for bending testing according to standard EN-14651 (apart from the specimen sizes)”

Line 163 - no need to appear again CMOD.

“displacement (CMOD), beam deflection and”

Line 166 - cut the word “and”

“In addition, the relationship between the specimen size, and the number and orientation of fibres in the fracture surface was also analysed.”

Sub-chapter 3.2

explain better "mould wall" – it is not clear. This characterization should be in point 2.1 Materials and specimen fabrication

Table 3 - The specimen presented in Table 3 are the same that are presented in Table 2?

it is unclear

Line 223 - “In addition, regular specimens prescribed by EN-14651 [27], with dimensions of 600x150x150mm³, also show improvements in the isotropic areas”

What specimen is this 600x150x150mmmm3? No test was reported during this work with this specimen; and it's also no citation of any other work done previously … Explain better or should be reported in point 2.1 Materials and specimen fabrication

Figure 8 and 10 should be smaller; more or less the same size as the rest

Author Response

This article deals with a very interesting experimental work, study the behaviour on beams (with three proportional sizes) and what the effects of the size effect in polyolefin fibre reinforced concrete under fracture testing. The paper provides new results and contributes to improve the state of the art in this line of research. It is considered that this paper can be published.

Is well written and easy to read.

Few improvements are needed. Please see the following comments aiming at improving the paper.

Figure 1. explain the he meaning LEFM?

 The authors have included the long term (linear elastic fracture mechanics) in the figure.

Figure 2. decrease the scale, for example 1/50

Given that the figure is parametric for the corresponding sizes, the authors have modified the figure in the sense suggested with the aim of adapting the size of the figures and the specimen to a more adequate scale.
Line 140 - what was the standard used for the manufacture of the specimens. In the test is referred that “The specimens were then ready for bending testing according to standard EN-14651 (apart from the specimen sizes)”

 The authors appreciate the comment and understand that some additional wording was needed in order to clarify the text. The manufacture of the specimens was performed by pouring concrete into the moulds from one side to the other in only one pour as it is commonly recommended for self-compacting fibre reinforced concrete (line 117 of the new text). Regarding the test procedures, the test method described by the standard was used even though the sizes were different. Both, the wording in the line 119 and line 144 have been clarified as follows:

“Fresh concrete was tested in a slump-flow test according to the standard procedures [25]. The double-test average results were t500= 6s and the diameter of the patty dm= 570 mm. Concretewas poured into the moulds from one end, leaving the concreteflow towards the opposite end with the only compaction action being its own weight as it is recommended by the standards and bibliography [23, 27]. The fresh specimens were covered with plastic film to keep the upper surface from drying. The beams were unmoulded 24 hours later and placed in a humid chamber at 20ºC and 90% of relative humidity for a minimum of 28 days (the point when they were ready for testing).

Before the bending tests, a half-depth cut was machined in the lower part of the beam central sections with a low-speed diamond disc, carefully maintaining the integrity and dimension of the upper half section of the specimen. The proportional dimensions based on the specimen depth D are shown in Figure 3. Under this procedure, three specimen size categories were obtained: large, medium and small, as defined in Table 2. Figure shows a photo with the three sizes of the tested specimens.

Figure 3.Specimen dimensional proportions and testing support and loading setup.

Table 2.Size and number of valid testing specimens.

Figure 4.Photo of the three sizes of the tested specimens.

2.2. Testing development

Once the specimens were ready for testing, the dimensions were checked in order to locate the positions of supports, loading cylinder and notching cuts. The surrounding areas of the cuts on both sides of the specimens were painted with a random point monochrome [26] template in order to apply digital image correlation (DIC) techniques. Crack generation and further development were detected by the position data of the template points during the test time. The specimens were then ready for bending testing according to standard EN-14651 (apart from the specimen sizes) [27]. That is to say, the test method in the standard was used to assess the fracture behaviour of different specimen sizes than that established by the standard (600x150x150mm³).The main testing equipment was one servo-control Instron 8803 press machine with a maximum loading capacity of 500 kN and a 25 kN Dynacell TM loading cell. A crack mouth opening displacement (CMOD) device, with a range of up to 8 mm, was used to measure the relative opening of the notching cut lower tips. The beam deflections were controlled by two linear variable differential transformer (LVDT) devices with 30 mm range, placed on either side of the beam. The final deflection value was the average of those of both devices. Furthermore, to verify the time development of the cracks two five-megapixel high-definition cameras were installed on either side of the testing arrangement for detecting and monitoring the displacements of points in the area where the cracks would presumably start. Recording frequency was set to one picture per second so that the pictures were synchronised with the testing machine.”

Line 163 - no need to appear again CMOD. 

“displacement (CMOD), beam deflection and”

 The authors have removed the acronym following the reviewer´s suggestion.

Line 166 - cut the word “and”

“In addition, the relationship between the specimen size, and the number and orientation of fibres in the fracture surface was also analysed.”

 The authors thank the reviewer for the correction and have removed the word.

Sub-chapter 3.2

explain better "mould wall" – it is not clear. This characterization should be in point 2.1 Materials and specimen fabrication

This explanation has been extended and have been shifted to the subsection 2.1, leaving in the subsection 3.2 the results of the fibre counting and the orientation factor and a picture of the fracture surfaces in Figure 6. All the figures have been renumbered accordingly. This wording has also been modified for clarification. The final wording moved to the subsection 2.1 is the following:

2.1 Materials and specimen fabrication

Portland cement type EN 197-1 CEM I 52.5 R-SR5 was used in the concrete mix called SCC10. The cement was mixed with limestone filler with a calcium carbonate content higher than 98%, a 45μm sieve retention of less than 0.05%, a density of 2700 kg/m3and a Blaine surface of 400-450 m2/kg. The silica aggregate contained two coarse size ranges: 4-8 mm grit and 8-12 mm gravel with a maximum size of 12.7 mm. The sand size range was 0-2 mm. Polycarboxilate-based superplasticizer, named Sika-Viscocrete 5720, with a density of 1090 kg/m³ and a solid content of 36% was also included in the mix. The mix formulation is shown in Table 1 and was that used in previous experimental campaigns [24]. The 48 mm long macro fibres type SikaFiber, with an embossed surface, were added to the mix in a volume fraction of 1.1%, equivalent to 10kg/m³. Table 1 also shows the main fibre properties. The visual aspect of the fibres can be seen in Figure 2.

Table 1.Concrete composition and fibre properties.

Figure 2.Visual aspect of the polyolefin fibres

A 100 litre capacity vertical axis mixer was used in the concrete fabrication. The mixing sequence was as follows. First, the three types of aggregate were mixed for one minute, followed by the addition of 1/3 of the fibre content which was mixed for 30 more seconds. Then, cement and filler were added and mixed for another 30 seconds. After that, 30 more mixing seconds were needed before the incorporation of the other 1/3 of fibre content.Next, after 75% of water had been poured and mixed for another minute the rest of the water, the fibres and the superplasticizer were added, continuing the mixing action for 150 seconds. Then the mixer was stopped for 150 seconds. The mixing was then resumed for two minutes.

Fresh concrete was tested in a slump-flow test according to the standard procedures [25]. The double-test average results were t500= 6s and the diameter of the patty dm= 570 mm. It was poured into the moulds from one end, leaving the concrete flow towards the opposite end with the only compaction action being its own weight as it is recommended by the standards and bibliography [23, 27]. The fresh specimens were covered with plastic film to keep the upper surface from drying. The beams were unmoulded 24 hours later and placed in a humid chamber at 20ºC and 90% of relative humidity for a minimum of 28 days (the point when they were ready for testing).

Before the bending tests, a half-depth cut was machined in the lower part of the beam central sections with a low-speed diamond disc, carefully maintaining the integrity and dimension of the upper half section of the specimen. The proportional dimensions based on the specimen depth D are shown in Figure 3. Under this procedure, three specimen size categories were obtained: large, medium and small, as defined in Table 2. Figure 4shows a photo with the three sizes of the tested specimens.

Figure 3.Specimen dimensional proportions and testing support and loading setup.

Table 2.Size and number of valid testing specimens.

Figure 4.Photo of the three sizes of the tested specimens.

2.2. Testing development

Once the specimens were ready for testing, the dimensions were checked in order to locate the positions of supports, loading cylinder and notching cuts. The surrounding areas of the cuts on both sides of the specimens were painted with a random point monochrome [26] template in order to apply digital image correlation (DIC) techniques. Crack generation and further development were detected by the position data of the template points during the test time. The specimens were then ready for bending testing according to standard EN-14651 (apart from the specimen sizes) [27]. That is to say, the test method in the standard was used to assess the fracture behaviour of different specimen sizes than that established by the standard (600x150x150mm³).The main testing equipment was one servo-control Instron 8803 press machine with a maximum loading capacity of 500 kN and a 25 kN Dynacell TM loading cell. A crack mouth opening displacement (CMOD) device, with a range of up to 8 mm, was used to measure the relative opening of the notching cut lower tips. The beam deflections were controlled by two linear variable differential transformer (LVDT) devices with 30 mm range, placed on either side of the beam. The final deflection value was the average of those of both devices. Furthermore, to verify the time development of the cracks two five-megapixel high-definition cameras were installed on either side of the testing arrangement for detecting and monitoring the displacements of points in the area where the cracks would presumably start. Recording frequency was set to one picture per second so that the pictures were synchronised with the testing machine.

The specimens were placed on the supports (as shown in Figure 3) and then the CMOD and LVDT devices were positioned by using a laser levelling instrument to ensure beam horizontality and that the loading cylinder was acting in the middle of the span (as shown in Figure 5). Lastly, the cameras were placed in position and activated on each side of the test configuration. 

The actuator displacement was used for loading control at an initial rate of 0.6 mm/min and in a second phase at 0.17 mm/min. In each test the first crack appeared in the first phase and the end of testing occurred in the second phase.

Figure 5Beam specimen ready for testing.

The positioning of fibres within the concrete matrix plays a significant role in the strength capacity of the lab specimen and the real structural element and its reliability [28]. A counting procedure of the existing fibres in the fracture surface was carried out based on dividing the section area in eight zones, as shown in Figure 6 in order to consider the main anisotropy effects such as the wall effect due to the formwork sizes. These zones are generated by using a band width of 24 mm (half of the fibre length) around the mould perimeter. The wall effect disturbs the fibre positioning and affects the fibre distribution and orientation. The wall effect tends to orientate the fibres along the surface of the wall and it is accepted in literature that it affects the surfaces closer than half the fibre length to the mould walls, as shown in Figure 6.

Figure 6. Mould wall effect and zones for fibre counting in the fracture surface.

The final wording of the subsection 3.2 can be read below:

3.2. Fracture surface analysis

In order to assess the number of fibres that cross the fracture surface, the specimens tested, shown in Table 2,weresplit in two half parts through the notch cut and the fibres counted in both parts. Figure 8 shows the two halves of one specimen prepared for the fibre counting.

Figure 8.Fracture surfaces of one specimen prepared for the fibre counting.

The orientation factor can be obtained by counting the number of fibres in the fracture surface and computing the theoretical (th) number of fibres that cross the section by means of equation (2). Such value is the number of fibres that would have been counted in an idealistic positioning.

The orientation factor (θ) can be defined as the relation of the number of fibres counted in a certain surface over its theoretical number. That is to say, it is possible to obtain the orientation factor once it has been counted by using the equation (3). 

with  being the cross section of the sample,  the section of one fibre,  the fibre volume fraction and n the number of fibres actually counted in situ.

In Table 3 the average numbers of fibres is shown, along with the variation coefficients and the global orientation factor, for each of the three specimen sizes. 

Table 3.Average number of fibres in the fracture surface with their coefficient of variation (c.v.) and the overall orientation factor (q) for each specimen size.

The orientation factor is a powerful tool for analysing and predicting reliability of structural use of FRC. In this case, as can be seen in Table 3, the orientation factor reached values that exceeded 0.60 (this was expected, given the results offered in the published literature [22, 28, 29]). The wall effect in the case of the small specimen affected more than half of the total fracture surface. Thus, this increased the value of qof the small specimens if compared with large and medium ones. In addition, standardspecimens prescribed by EN-14651 [27], with dimensions of 600x150x150mm³,also show improvements in the isotropic areas given that the flux of self-compacting concrete tends to align the fibres perpendicular to the fracture surface. This helps understanding of why qis greater than 0.60. Moreover, previous research [30, 31] showed that the longer the pouring distance ofself-compacting concrete, the higher is the influence of the flux. That is to say, the values of the orientation factor of the large specimens were expected to be higher than the medium ones. In addition, the wall effect was evident on the orientation factor of the small specimens, with all of them being higher than a regular specimen size with a regular vibrated concrete. 

Table 3 - The specimen presented in Table 3 are the same that are presented in Table 2? 

it is unclear

In order to clarify this aspect, the wording has modified as follows:

 In order to assess the number of fibres that cross the fracture surface, the specimenstested, shown in Table 2,weresplit in two half parts through the notch cut and the fibres counted in both parts. In Table 3 the average numbers of fibres is shown, along with the variation coefficients and the global orientation factor, for each of the three specimen sizes. 

Line 223 - “In addition, regular specimens prescribed by EN-14651 [27], with dimensions of 600x150x150mm³, also show improvements in the isotropic areas”

What specimen is this 600x150x150mmmm3? No test was reported during this work with this specimen; and it's also no citation of any other work done previously … Explain better or should be reported in point 2.1 Materials and specimen fabrication

 The authors comprehend that this point should be clarified. The standard size set in EN-14651 is 600x150x150mm³. This work deals with size effect and, thus, other sizes have been used in order to assess the size effect of this material. In order to ease the reading and avoid confusion of size, some wording has been modified and removed. This modifications can be read below:

“The orientation factor is a powerful tool for analysing and predicting reliability of structural use of FRC. In this case, as can be seen in Table 3, the orientation factor reached values that exceeded 0.60 (this was expected, given the results offered in the published literature [22, 28, 29]). The wall effect in the case of the small specimen affected more than half of the total fracture surface. Thus, this increased the value of qof the small specimens if compared with large and medium ones. In addition, regularstandardspecimens prescribed by EN-14651 [27], with dimensions of 600x150x150mm³,also show improvements in the isotropic areas given that the flux of self-compacting concrete tends to align the fibres perpendicular to the fracture surface. This helps understanding of why qis greater than 0.60. Moreover, previous research [30, 31] showed that the longer the pouring distance ofself-compacting concrete, the higher is the influence of the flux. That is to say, the values of the orientation factor of the large specimens were expected to be higher than the medium ones. In addition, the wall effect was evident on the orientation factor of the small specimens, with all of them being higher than a regular specimen size with a regular vibrated concrete.” 

Figure 8 and 10 should be smaller; more or less the same size as the rest

The size of the figures have been modified in order to be in similar sizes compared with the other figures.

Submission Date

15 March 2019

Date of this review

01 Apr 2019 17:19:04

Reviewer 2 Report

1) Why only three sizes are chosen? How do you expect the results to change of sizes go 1 order of magnitude lower/higher than the current lowest/highest sizes.

2) How do you anticipate the effects of three different sizes of PP fibers used? Will results seem have any effect of fiber size? What cab be the effect of fiber size/aspect ratio distribution?

3) Why the authors did not consider conducting experiments without any fibers to study the effect of fiber.

Author Response

1) Why only three sizes are chosen? How do you expect the results to change of sizes go 1 order of magnitude lower/higher than the current lowest/highest sizes.

The authors agree with the reviewer about the interest of studying different sizes of the specimens, from very small (in the limit of the homogeneity of the materials: two/three times of the maximum size of the aggregate) to very large ones (size of a dam). In order to obtain experimental values in a lab, the three proportional sizes adopted are used in concordance with previous works about size effect in plain concrete (32-33, among others). Since the paper is based on experimental data, the authors think that the extrapolation to larger specimens by applying analytical formulae is out of the scope of the paper.

33. Guinea, G.V , Plana, J. Elices, M. “”Correlation between the softening and the size effect curves”. In “size Effect in Concrete Structures”, H. Mihashi, H. Okamura and Z.P Bazant, E&F Spon, London, pp.233-244, 1994.

In any event, in order to clarify this point, some wording has been added as follows:

Regarding such an effect, it is important to note that there are three theories involved in its study: statistical, deterministic and that based on fractals [15], with the deterministic one being the most used [16]. The size effect is defined as the deviation of the real strength capacity compared with the load predicted by plastic analysis or another classical structural strength theory based on critical stress states [17]. This effect can be graphically represented, as in Figure 1, being at least three sizes necessary in order to fit the size effect law of the material.

 2) How do you anticipate the effects of three different sizes of PP fibers used? Will results seem have any effect of fiber size? What cab be the effect of fiber size/aspect ratio distribution?

This is such an interesting and valuable question. The influence of the fibre sizes as well as the aggregates can be of relative importance and also in relation with the mould sizes. Some aspects dealing with these feature have been addressed in previous research and are cited in references [30, 31]. With such variations, not only the effectiveness of the fibres can show relevant variations but also the fibre positioning and orientation factor can vary significantly. In addition, the areas influenced by the wall effect would vary. In references [30, 31] the authors found that even some improvements with self-compacting concrete were found for 48mm long polyolefin fibres due to the fewer number of fibres with enough embedded length present if compared with 60mm long fibres. Hence, although fracture results were very similar, the resisting mechanisms were different. Given that this is such a complex question with a large amount of variables, the authors decided to keep fibre length as a constant in this work. Although this matter was out of the scope of this article the authors will continue working on understanding this matter in future research.

3) Why the authors did not consider conducting experiments without any fibers to study the effect of fiber.

The authors understand that in some works the use of a plain concrete as a reference is of interest. Nevertheless, size effect in plain concrete has been thoroughly studied in previous references and the main objective of the research was to analyse how the size effect affects fibre reinforced concrete and why there was no consensus in literature about the influence of the element size in the residual strengths. As can be seen in the results for fLOP(see new Figure 10)itsuch value is governed by the concrete matrix properties (see ref. 33) and the size effect correspond with those expected for plain concrete.

Submission Date

15 March 2019

Date of this review

02 Apr 2019 05:28:20

Reviewer 3 Report

In this paper, the size effect on flexural fracture of polyolefin fiber reinterpreted concrete (PFRC) was investigated. And the results can contribute to the development of research on flexural behavior of fiber reinforced concrete. However, I believe that proper modification of the following are necessary for the publication of this paper.

1. Does the volume of the fiber include in the volume of the concrete? It is necessary to present clearly.

2. Presenting pictures of the used fibers can help the reader understand.

3. Please provide a detailed description and photo of the fiber number in the matrix and how to obtain an orientation factor. More detail is needed because it is an important factor in the analysis of the results.

4. In this paper, the length and height of the specimen have changed to examine the size effect. If there is a reason why the width of the test specimen is the same, add it.

5. Table 2 shows that two medium-sized specimens are used. Is it a wrong write? If not, please give me a reason.

6. Adding a detailed description of the digital image correction (DIC) techniques and pictures of the installation situation can help the reader understanding it easily.

7. For the analysis shown in Figure 10, it is difficult to understand the comparison of the size effects under the conditions of the same number of fibers. Because fibers are mixed in volume fraction, it is understood that if the size of the fracture surface differs, the fiber volume fraction will be reduced. I think it is not a size effect, it is an effect of fiber volume fraction. Please give me your opinion.

8. Please provide a formula for obtaining fracture energy or related standards.

9. In Figure 11, an additional picture is required to show the differences in the size of the specimen. In addition, please add the results of the image analysis to the size effects covered in this study.

Author Response

In this paper, the size effect on flexural fracture of polyolefin fiber reinterpreted concrete (PFRC) was investigated. And the results can contribute to the development of research on flexural behavior of fiber reinforced concrete. However, I believe that proper modification of the following are necessary for the publication of this paper.

1. Does the volume of the fiber include in the volume of the concrete? It is necessary to present clearly. 

The volume fraction of fibres have been included in Table 1:

2. Presenting pictures of the used fibers can help the reader understand. 

In order to ease the comprehension of the wording, a new figure (Figure 2) has been included and all the figures renumbered. 

Figure 2.Visual aspect of the polyolefin fibres

3. Please provide a detailed description and photo of the fiber number in the matrix and how to obtain an orientation factor. More detail is needed because it is an important factor in the analysis of the results.

The authors understand that this is an important issue in this article. Therefore, they have included a new figure (Figure 8) as well as new wording in order to explain how the orientation factor can be computed. The final wording is the following:

In order to assess the number of fibres that cross the fracture surface, the specimens tested, shown in Table 2,weresplit in two half parts through the notch cut and the fibres counted in both parts. Figure 7 shows the two halves of one specimen prepared for the fibre counting.

Figure 8.Fracture surfaces of one specimen prepared for the fibre counting.

The orientation factor can be obtained by counting the number of fibres in the fracture surface and computing the theoretical (th ) number of fibres that cross the section by means of expression (2). Such value is the number of fibres that would have been counted in an idealistic positioning.

The orientation factor (θ) can be defined as the relation of the number of fibres counted in a certain surface over its theoretical number. That is to say, it is possible to obtain the orientation factor once it has been counted by using the expression (3). 

with  being the cross section of the sample,  the section of one fibre,  the fibre volume fraction and n the number of fibres actually counted in situ.

4. In this paper, the length and height of the specimen have changed to examine the size effect. If there is a reason why the width of the test specimen is the same, add it.

The authors believe that the optimum size effect study would be performed by modifying the three dimensions parametrically. However, as a matter of time and resources in a construction materials laboratory, the biggest size would become unapproachable in terms of weight for testing. Moreover, in terms of residual strengths and concrete fracture mechanics the thickness would have negligible influence. Having said that, it is true that with fibre reinforced concrete the thickness could imply a two-dimensional fibre positioning with slight variations in the three sizes given the wall effect. This study is remarkably deep in such sense and the authors think that the results show the fracture behaviour of the three sizes in a representative manner and allow supporting both discussion and conclusion section which are significantly novel.

5. Table 2 shows that two medium-sized specimens are used. Is it a wrong write? If not, please give me a reason. 

The authors understand the comment and would like to explain the reasons of the use of only two specimens in the intermediate size. Three specimens of each size were produced. However, in the positioning of one of the medium-sized specimens for the test, it became pre-cracked and therefore this results were not considered for the analysis. Moreover, as the two results obtained showed hardly any scattering between them, the behaviour of the material at medium sizes was considered statistically valid. 

6. Adding a detailed description of the digital image correction (DIC) techniques and pictures of the installation situation can help the reader understanding it easily.

The authors think that including a new figure and some additional wording following the suggestion will improve the paper. Hence, they have included Figure 13 and have explained in more detail how DIC can be used. The final wording is the following:

4.4. Video-extensometry

The generation and development of cracks were studied by means of digital image correlation (DIC). That is to say, a correlation of images synchronised with the testing machine data record allows the use of the images as a video-extensometer. In order to perform the DIC analyses, the images were recorded at one frame per second. Such a frame rate allowed the synchronisation of the videos and the data acquired by testing machine gauges. As previously mentioned, the testing machine recorded load, displacement of the actuator, CMOD in both sides and test elapsed time. Therefore, time was common for both systems and the fixed frame rate allowed the synchronisation of the videos with the testing machine and made it possible to correlate by using the software Vic-2D.The scheme of the test and disposition of the devices can be seen in Figure 13.This study allowed the relationship of the load values produced by the actuator displacement with the crack formation images at a given test elapsed time to be established. In each case, the crack generation started in the upper point of the notch cut. The cracks progressed with a vertical trajectory clearly influenced by the fibre bridging effect. Figure 14is the image of one of the cracks generated in the bending test. 

Figure 13.Crack generated in one of the specimens during the bending test.Bending test setup including DIC positioning.

Figure 14.Crack generated in one of the specimens during the bending test.

7. For the analysis shown in Figure 10, it is difficult to understand the comparison of the size effects under the conditions of the same number of fibers. Because fibers are mixed in volume fraction, it is understood that if the size of the fracture surface differs, the fiber volume fraction will be reduced. I think it is not a size effect, it is an effect of fiber volume fraction. Please give me your opinion.

The authors believe that it is important to clarify this issue and have included this new wording:

Figure 11 shows the values of fR1, fR3, fminand fREMas a function of the number of fibres in the fracture surface. As can be seen, keeping constant the volume fraction, the number of fibres in the fracture surface in the smallest specimens is about 50, in the medium-sized specimens around 100 and close to 200 for the largest specimens. In order to assess the size effect in fmin, Figure 12 was performed. In this figure, the results of Figure 11(a) were extended by a linear fitting for the comparison with the same number of fibres. As it has been shown by reference [34] the residual strength in fibre reinforced concrete is directly related with the number of fibres in the fracture surface. Figure 12 shows that with the same number of fibres there is a clear size effect. Considering, for example, 50 fibres the strength of the large specimens (point A) corresponds to a minimum value of 0.48 MPa, whereas this value increases to 0.78 MPa (point B) for medium size specimens and reaches 1.11 MPa (point C) in case of small specimens. This data means a 231% strength increment in the small specimens in relation to the big specimens. Again, this behaviour confirmed that the size effect occurred for PFRC elements, according to the classical theory of size effect for quasi-brittle materials [32].

8. Please provide a formula for obtaining fracture energy or related standards. 

The authors have included the following explanation and formula in the beginning of subsection 4.2:

4.2. Fracture energy

In order to quantify and compare the work of fracture for each specimen size, the specific fracture energy (GF) of all the mixes was analysed by means of the load-deflection curves and processed by using equation (4).

Where Wfwas the fracture work borne by the sample, bthe width of the sample (50mm) and hspthe length of the ligament (0.5D for each size).

9. In Figure 11, an additional picture is required to show the differences in the size of the specimen. In addition, please add the results of the image analysis to the size effects covered in this study.

The authors think that Figure 3 (now Figure 4) shows three specimens after the test so that the differences in the size can be adequately perceived. In any case, they have no objection in including new pictures if required. 

As for the use of DIC, the authors believe that this technique might be of interest for future research works though it does not provide with any additional information of interest regarding with the size effect. Thus, they believe that including additional data would not improve the quality of the paper. 

Submission Date

15 March 2019

Date of this review

02 Apr 2019 07:18:51

Round  2

Reviewer 2 Report

Thank you for the responses. 

Reviewer 3 Report

Thank you for your efforts in revising your paper. 
I think the revised paper has been revised to reflect the review comments appropriately. 
I recommend that this paper be published in the journal.